# OMNIPARSER FOR PURE VISION BASED GUI AGENT

## ABSTRACT

The recent advancements of large vision language models shows their great potential in driving the agent system operating on user interfaces. However, we argue that the power multimodal models like GPT-4V as a general agent on multiple operating systems across different applications is largely underestimated due to the lack of a robust screen parsing technique capable of: 1) reliably identifying interactable icons within the user interface, and 2) understanding the semantics of various elements in a screenshot and accurately associate the intended action with the corresponding region on the screen. To fill these gaps, we introduce OMNI-PARSER, a comprehensive method for parsing general user interface screenshots into structured elements, which significantly enhances the ability of GPT-4V to generate actions that can be accurately grounded in the corresponding regions of the interface. We first curated an interactable icon detection dataset using popular webpages and an icon description dataset. These datasets were utilized to fine-tune specialized models: a detection model to parse interactable regions on the screen and a caption model to extract the functional semantics of the detected elements. OMNIPARSER significantly improves GPT-4V's performance on ScreenSpot benchmark. And on Mind2Web and AITW benchmark, OMNIPARSER with screenshot only input outperforms the GPT-4V baselines requiring additional information outside of screenshot. We further demonstrate that OMNIPARSER can seamlessly integrate with other vision language models, significantly enhancing their agentic capabilities.

## 1 INTRODUCTION

Large language models have shown great success in their understanding and reasoning capabilities. More recent works have explored the use of large vision-language models (VLMs) as agents to perform complex tasks on the user interface (UI) with the aim of completing tedious tasks to replace human efforts (YZL[+]23; YYZ[+]23; DGZ[+]23; ZGK[+]24; HWL[+]23; YZS[+]24; WXJ[+]24; GFH[+]24; CSC[+]24). Despite the promising results, there remains a significant gap between current state-of-the-arts and creating widely usable agents that can work across multiple platforms, *e.g.* Windows/MacOS, IOS/Android and multiple applications (Web broswer Office365, PhotoShop, Adobe), with most previous work focusing on limiting applications or platforms.

While large multimodal models like GPT-4V and other models trained on UI data (HWL[+]23; YZS[+]24; CSC[+]24) have demonstrated abilities to understand basic elements of the UI screenshot, action grounding remains one of the key challenges in converting the actions predicted by LLMs to the actual actions on screen in terms of keyboard/mouse movement or API call (ZGK[+]24). It has been noted that GPT-4V is unable to produce the exact x-y coordinate of the button location, Set-of-Mark prompting (YZL[+]23) proposes to overlay a group of bounding boxes each with unique numeric IDs on to the original image, as a visual prompt sent to the GPT-4V model. By applying set-of-marks prompting, GPT-4V is able to ground the action into a specific bounding box which has ground truth location instead of a specific xy coordinate value, which greatly improves the robustness of the action grounding (ZGK[+]24). However, the current solutions using SoM relies on parsed HTML information to extract bounding boxes for actionable elements such as buttons, which limits its usage to web browsing tasks. We aim to build a general approach that works on a variety of platforms and applications.

In this work, we argue that previous pure vision-based screen parsing techniques are not satisfactory, which lead to significant underestimation of GPT-4V model's understanding capabilities. And a

reliable vision-based screen parsing method that works well on general user interface is a key to improve the robustness of the agentic workflow on various operating systems and applications. We present OMNIPARSER, a general screen parsing tool to extract information from UI screenshot into structured bounding box and labels which enhances GPT-4V's performance in action prediction in a variety of user tasks.

We summarize our contributions as follows:

- We curate a interactable region detection dataset using bounding boxes extracted from DOM tree of popular webpages.

- We propose OmniParser, a pure vision-based user interface screen parsing method that combines multiple finetuned models for better screen understanding and easier grounded action generation.

- We evaluate OmniParser on the ScreenSpot, Mind2Web, and AITW benchmarks, demonstrating significant improvement over the GPT-4V baseline, using only screenshots as input.

- We show that OMNIPARSER operates seamlessly and serves as an easy-to-integrate tool for a variety of state-of-the-art public vision language models. And we open-source both our code and model to facilitate further development and integration of OMNIPARSER to other vision language models. [1]

## 2 RELATED WORKS

### 2.1 UI SCREEN UNDERSTANDING

There has been a line of modeling works focusing on detailed understanding of UI screens, such as Screen2Words (WLZ[+]21), UI-BERT (BZX[+]21), WidgetCaptioning (LLH[+]20), Action-BERT (HSZ[+]21). These works demonstrated effective usage of multimodal models for extracting semantics of user screen. But these models rely on additional information such as view hierarchy, or are trained for visual question answering tasks or screen summary tasks.

There are also a couple publicly available datasets on UI screen understanding. Most notably the Rico dataset (DHF[+]17), which contains more than 66k unique UI screens and its view hierarchies. Later (SWL[+]22) auguments Rico by providing 500k human annotations on the original 66k RICO screens identifying various icons based on their shapes and semantics, and associations between selected general UI elements (like icons, form fields, radio buttons, text inputs) and their text labels. Same on mobile platform, PixelHelp (LHZ[+]20) provides a dataset that contains UI elements of screen spanning across 88 common tasks. In the same paper they also released RicoSCA which is a cleaned version of Rico. For the web and general OS domain, there are several works such Mind2Web (DGZ[+]23), MiniWob++(LGP[+]18), Visual-WebArena (KLJ[+]24; ZXZ[+]24), and OS-World (XZC[+]24) that provide simulated environment, but does not provide dataset explicitly for general screen understanding tasks such as interactable icon detection on real world websites.

To address the absence of a large-scale, general web UI understanding dataset, and to keep pace with the rapid evolution of UI design, we curated an icon detection dataset using the DOM information from popular URLs avaialbe on the Web. This dataset features the up-to-date design of icons and buttons, with their bounding boxes retrieved from the DOM tree, providing ground truth locations.

### 2.2 AUTONOMOUS GUI AGENT

Recently there has been a lot of works on designing autonomous GUI agent to perform tasks in place of human users. One line of work is to train an end-to-end model to directly predict the next action, representative works include: Pixel2Act (SJC[+]23), WebGUM(FLN[+]24) in web domain, Ferret (YZS[+]24), CogAgent (HWL[+]23), and Fuyu (BEH[+]23) in Mobile domain. Another line of works involve leveraging existing multimodal models such as GPT-4V to perform user tasks. Representative works include MindAct agent (DGZ[+]23), SeeAct agent (ZGK[+]24) in web domain and agents in (YYZ[+]23; WXY[+]24; RLR[+]23) for mobile domain. These work often leverages the DOM information in web browser, or the view hierarchies in mobile apps to get the ground

---

[1]Github repository and Huggingface model links will be made available after the review.

truth position of interactable elements of the screen, and use Set-Of-Marks(YZL[+]23) to overlay the bounding boxes on top of the screenshot then feed into the vision-language models. However, ground truth information of interactable elements may not always be available when the goal is to build a general agent for cross-platforms and cross-applications tasks. Therefore, we focus on providing a systematic approach for getting structured elements from general user screens.

# 3 METHODS

To complete a complex task, $T$ on graphical user interface, the process can usually be broken down into several steps of state-action pairs $(S_0, A_0), ..., (S_n, A_n)$, where $n$ is the number of steps. Each step requires the model's (e.g. GPT-4V) ability to: 1) understand the state of the UI screen information $S_i$ in the current step, i.e. analyzing what is the screen content overall, what are the functions of detected icons that are labeled with numeric ID, and 2) predict what is the next action $A_i$ on the current screen that is likely to help completing the whole task. Normally this can be formulated as:

$$a_{i+1} = \pi(T, S_i, [(S_0, A_0), ..., (S_{i-1}, A_{i-1})]$$
$$S_i = \{h_i, \text{Img}_i\} \tag{1}$$

Here $S_i$ encapsules the current information of the screen, which varies according to the model $\pi$'s input types and the operating environment. For example, in the web tasks (ZGK[+]24), $S_i = \{h_i, \text{Img}_i\}$, where $h_i$ is the HTML information at step $i$, and $\text{Img}_i$ is the screenshot of the GUI at step $i$. In Android tasks (RLR[+]23), $h_i$ is the view hierarchy information. In these cases, the model $\pi$ is required to extract information directly from $\text{Img}_i$ and at the same time generate an action prediction. Instead of trying to accomplish the two goals in one call, we found it beneficial to extract some of the information such as semantics in the screen parsing stage, to alleviate the burden of GPT-4V so that it can leverages more information from the parsed screen and focus more on the action prediction.

Hence we propose OMNIPARSER, which integrates the outputs from a finetuned interactable icon detection model, a finetuned icon description model, and an OCR module. This combination produces a structured state informaton $S_i$, which includes a DOM-like representation of the UI and a screenshot overlaid with bounding boxes for potential interactable elements

$$a_{i+1} = \pi(T, S_i, [(S_0, A_0), ..., (S_{i-1}, A_{i-1})]$$
$$S_i = \{\text{Img}_i^{som}, \text{LS}_i, \text{OCR}_i^{txt}\} \tag{2}$$
$$\text{Img}_i^{som}, \text{LS}_i, \text{OCR}_i^{txt} = \text{OMNIPARSER}(\text{Img}_i)$$

Here $\text{Img}_i^{som}$ is the set-of-mark image labeled by the finetuned interactable icon detection model, $\text{LS}_i$ is the local semantics output by the finetuned icon description model. We discuss each component of the OMNIPARSER in more details for the rest of the section.

## 3.1 INTERACTABLE REGION DETECTION

Identifying interactable regions from the UI screen is a crucial step to reason about what actions should be performed given a user tasks. Instead of directly prompting GPT-4V to predict the xy coordinate value of the screen that it should operate on, we follow previous works to use the Set-of-Marks approach (YZL[+]23) to overlay bounding boxes of interactable icons on top of UI screenshot, and ask GPT-4V to generate the bounding box ID to perform action on. However, different from (ZGK[+]24; KLJ[+]24) which uses the ground truth button location retrieved from DOM tree in web browswer, and (YYZ[+]23) which uses labeled bounding boxes in the AITW dataset (RLR[+]23), we finetune a detection model to extract interactable icons/buttons.

Specifically, we curate a dataset of interactable icon detection dataset, containing 67k unique screenshot images, each labeled with bounding boxes of interactable icons derived from DOM tree. We first took a 100k uniform sample of popular publicly availabe urls on the web (OXL[+]22), and collect bounding boxes of interactable regions of the webpage from the DOM tree of each urls. Some examples of the webpage and the interactable regions are shown in 2.

Apart from interactable region detection, we also have a OCR module to extract bounding boxes of texts. Then we merge the bounding boxes from OCR detection module and icon detection module

while removing the boxes that have high overlap (we use 90% as a threshold). For every bounding box, we label it with a unique ID next to it using a simple algorithm to minimizing the overlap between numeric labels and other bounding boxes.

## 3.2 Incorporating Local Semantics of Functionality

We found in a lot of cases where only inputting the UI screenshot overlayed with bounding boxes and associated IDs can be misleading to GPT-4V. We argue the limitation stems from GPT-4V's constrained ability to simultaneously perform the composite tasks of identifying each icon's semantic information and predicting the next action on a specific icon box. This has also been observed by several other works (YYZ[+]23; ZGK[+]24). To address this issue, we incorporate the local semantics of functionality into the prompt, i.e. for each icons detected by the interactable region detection model, we use a finetuned model to generate description of functionality to the icons, and for each text boxes, we use the detected texts and its label.

We perform more detailed analysis for this topic in section 4.1. To the best of our knowledge, there is no public model that is specifically trained for up-to-date UI icon description, and is suitable for our purpose to provide fast and accurate local semantics for the UI screenshot. Therefore we curate a dataset of 7k icon-description pairs using GPT-4o, and finetune a BLIP-v2 model (LLSH23) on this dataset. Details of dataset and training can be found in Appendix 8.1. After finetuning, we found the model is much more reliable in its description to common app icons. Examples can be seen in figure 5. And in figure 3, we show it is helpful to incorporate the semantics of local bounding boxes in the form of text prompt along with the UI screenshot visual prompt.

## 4 Experiments and Results

We conduct experiments on several benchmarks to demonstrate the effectiveness of OMNIPARSER. We start by a motivating experiments showing that current GPT-4V model with set of mark prompting (YZL[+]23) is prone to incorrectly assigning label ID to the referred bounding boxes. Then we evaluate on Seeclick benchmark and Mind2Web to further showcase OMNIPARSER with local semantics can improve the GPT-4V's performance on real user tasks on different platforms and applications.

## 4.1 Evaluation on SeeAssign Task

To test the ability of correctly predicting the label ID given the description of the bounding boxes for GPT-4v models, We handcrafted a dataset SeeAssign that contains 112 tasks consisting of samples from 3 different platforms: Mobile, Desktop and Web Browser. Each task includes a concise task description and a screenshot image. The task descriptions are manually created and we make sure each task refers to one of the detected bounding boxes, e.g. 'click on 'settings'', 'click on the minimize button'. During evaluation, GPT-4V is prompted to predict the bounding box ID associated to it. Detailed prompt are specified in Appendix. The task screenshot images are sampled from the ScreenSpot (CSC[+]24) benchmark, where they are labeled with set of marks using OMNIPARSER. The tasks are further divided into 3 sub-categories by difficulty: easy (less than 10 bounding boxes), medium (10-40 bounding boxes) and hard (more than 40 bounding boxes).

From table 1, we see that GPT-4V often mistakenly assign the numeric ID to the table especially when there are a lot of bounding boxes over the screen. And by adding local semantics including texts within the boxes and short descriptions of the detected icons, GPT-4V's ability of correctly assigning the icon improves from 0.705 to 0.938.

From figure 3, we see that without the description of the referred icon in the task, GPT-4V often fails to link the icon required in the task and the ground truth icon ID in the SoM labeled screenshot, which leads to hallucination in the response. With fine-grain local semantics added in the text prompt, it makes it much easier for GPT-4V to find the correct icon ID for the referred icon.

## 4.2 Evaluation on ScreenSpot

ScreenSpot dataset (CSC[+]24) is a benchmark dataset that includes over 600 interface screenshots from mobile (iOS, Android), desktop (macOS, Windows), and web platforms. The task instructions

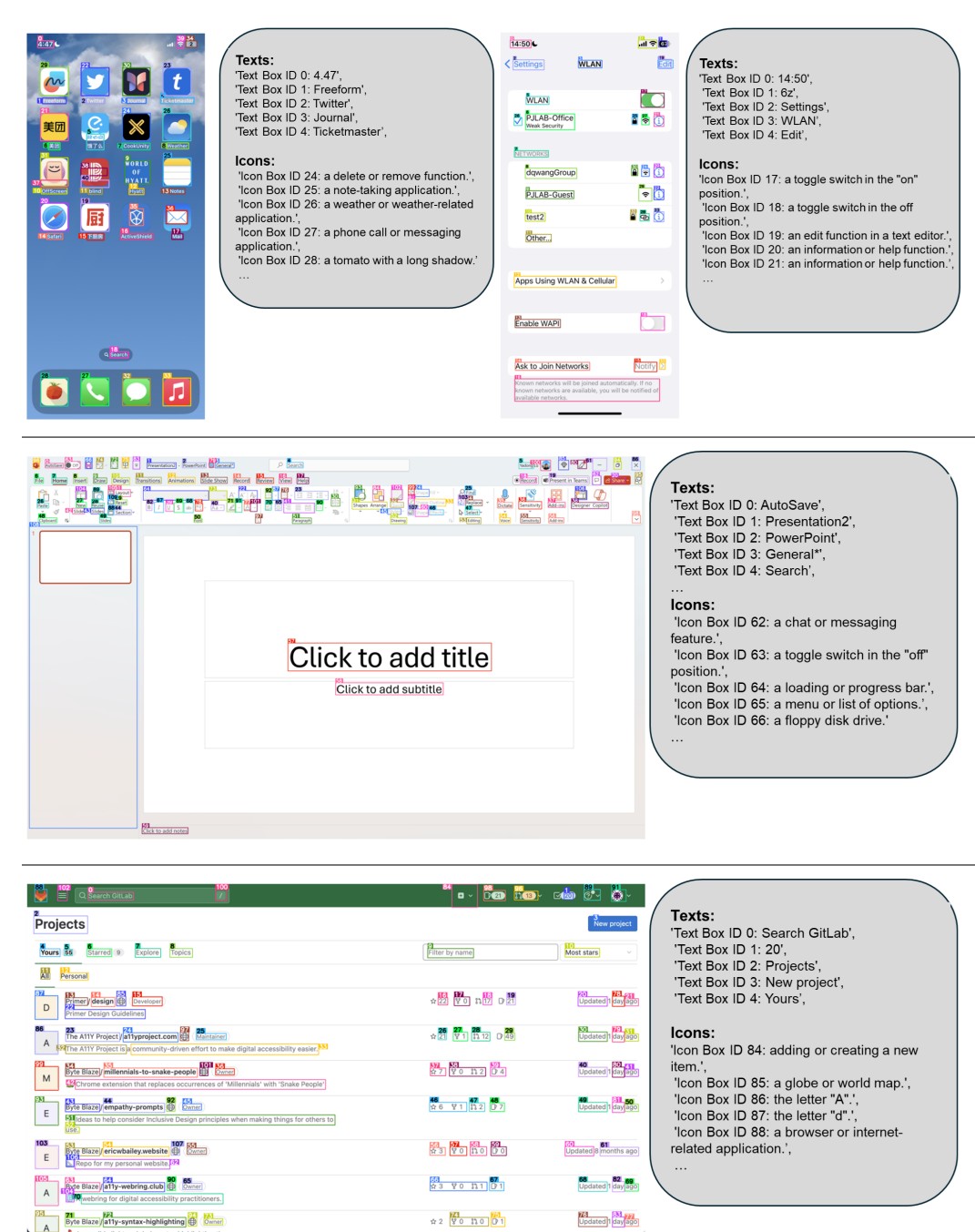

Figure 1: Examples of parsed screenshot image and local semantics by OMNIPARSER. The inputs to OmniParse are user task and UI screenshot, from which it will produce: 1) parsed screenshot image with bounding boxes and numeric IDs overlayed, and 2) local semantics containing both text extracted and icon description.

are manually created so that each instruction corresponds to an actionable elements on the UI screen. We first evaluate the performance of OMNIPARSER using the this benchmark. In table 2, we can see across the 3 different platforms: Mobile, Desktop and Web, OMNIPARSER significantly improves the GPT-4V baseline from 16.2% to 73.0%. Noticeably, OMNIPARSER's performance even surpasses models that are specifically finetuned on GUI dataset including SeeClick, CogAgent and Fuyu by a large margin. We also note that incorporating the local semantics (OMNIPARSER w. LS in the

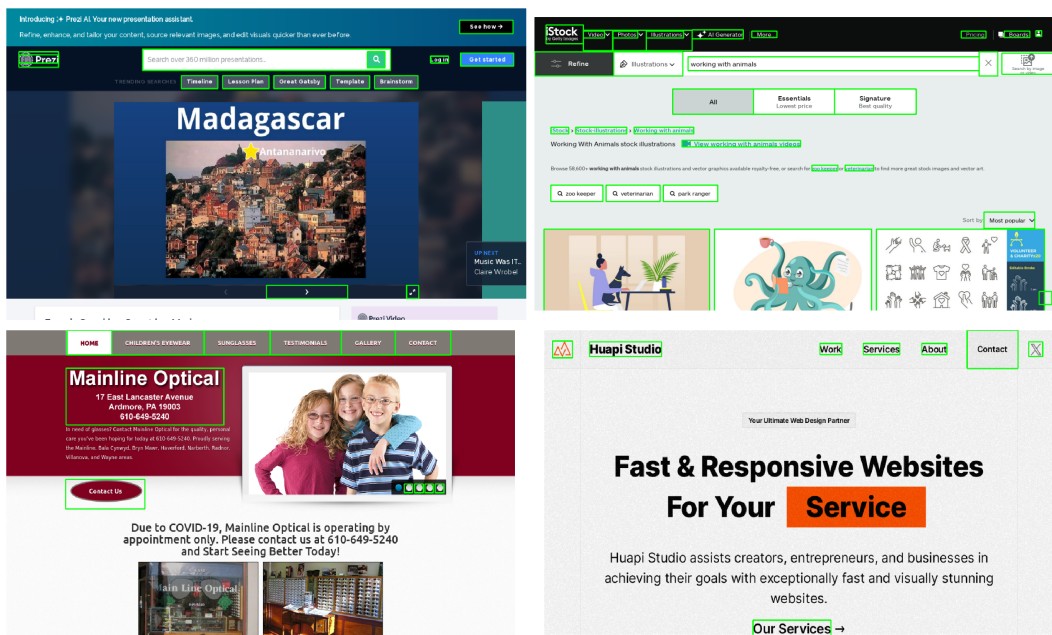

Figure 2: Examples from the Interactable Region Detection dataset. The bounding boxes are based on the interactable region extracted from the DOM tree of the webpage.

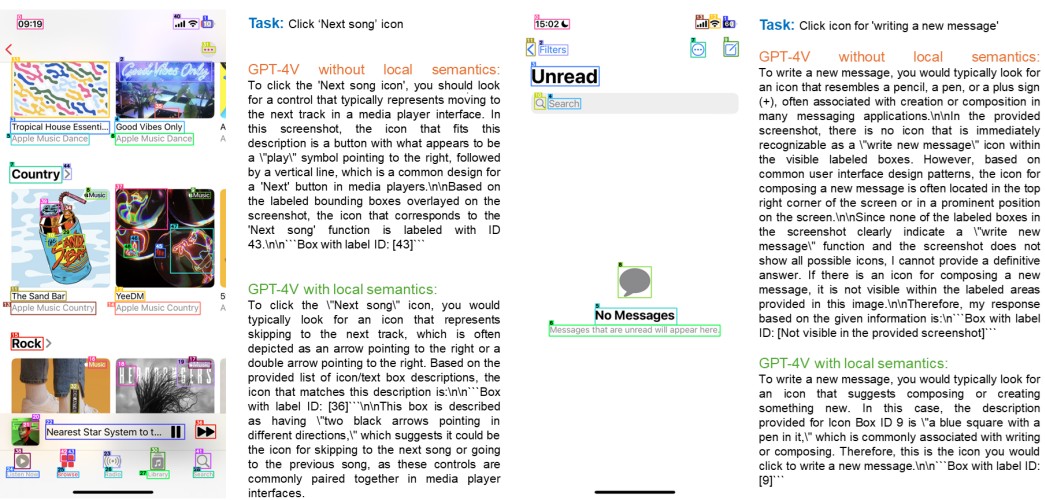

Figure 3: Examples from the SeeAssign evaluation. We can see that fine-grain local semantics improves the GPT-4V's ability to assign correct labels to the referred icon.

table) further improves the overall performance. This corroborates with the finds in section 4.1 that incorporating local semantics of the UI screenshot in text format, i.e. adding OCR text and descriptions of the icon bounding boxes further helps GPT-4V to accurately identify the correct element to operate on. Furthermore, our findings indicate that the interactable region detection (ID) model we finetuned improves overall accuracy by an additional 4.3% compared to using the raw Grounding DINO model. This underscores the importance of accurately detecting interactable elements for the success of UI tasks. Overall, the results demonstrate that the UI screen understanding capability of GPT-4V is significantly underestimated and can be greatly enhanced with more accurate interactable elements detection and the incorporation of functional local semantics. More ablations across other open-source vision language models can be found in section 5.

|  | Easy | Medium | Hard | Overall |
|---|---|---|---|---|
| GPT-4V w.o. local semantics | 0.913 | 0.692 | 0.620 | 0.705 |
| GPT-4V w. local semantics | 1.00 | 0.949 | 0.900 | 0.938 |

Table 1: Comparison of GPT-4V with and without local semantics

| Methods | Model Size | Mobile | | Desktop | | Web | | Average |
|---|---|---|---|---|---|---|---|---|
| | | Text | Icon/Widget | Text | Icon/Widget | Text | Icon/Widget | |
| Fuyu | 8B | 41.0% | 1.3% | 33.0% | 3.6% | 33.9% | 4.4% | 19.5% |
| CogAgent | 18B | 67.0% | 24.0% | 74.2% | 20.0% | 70.4% | 28.6% | 47.4% |
| SeeClick | 9.6B | 78.0% | 52.0% | 72.2% | 30.0% | 55.7% | 32.5% | 53.4% |
| MiniGPT-v2 | 7B | 8.4% | 6.6% | 6.2% | 2.9% | 6.5% | 3.4% | 5.7% |
| Qwen-VL | 9.6B | 9.5% | 4.8% | 5.7% | 5.0% | 3.5% | 2.4% | 5.2% |
| GPT-4V | - | 22.6% | 24.5% | 20.2% | 11.8% | 9.2% | 8.8% | 16.2% |
| OmniParser (GPT-4V) | - | 93.9% | **57.0%** | **91.3%** | **63.6%** | 81.3 | **51.0%** | **73.0%** |
| - w.o. ID | - | **94.8%** | 53.7% | 89.3% | 44.9% | **83.0%** | 45.1% | 68.7% |
| - w.o. ID and w.o. LS | - | 92.7% | 49.4% | 64.9% | 26.3% | 77.3% | 39.7% | 58.38% |

Table 2: Comparison of different approaches on ScreenSpot Benchmark. LS is short for local semantics of functionality and ID is short for the interactable region detection model we finetune. The setting w.o. ID means we replace the ID model with original Grounding DINO model not finetuned on our data, and with local semantics. The setting w.o. ID and w.o LS means we use Grounding DINO model, and further without using the icon description in the text prompt.

## 4.3 EVALUATION ON MIND2WEB

In order to test how OMNIPARSER is helpful to the web navigation secnario, We evaluate on (DGZ+23) benchmark. There are 3 different categories of task in the test set: Cross-Domain, Cross-Website, and Cross-Tasks. We used a cleaned version of Mind2Web tests set processed from the raw HTML dump which eliminates a small number of samples that has incorrect bounding boxes. In total we have 867, 167, 242 tasks in the test set from Cross-Domain, Cross-Website, and Cross-Tasks category respectively. During evaluation, we feed both the parsed screen results and the action history as text prompt, and SOM labeled screenshot to GPT-4V similar to the prompting strategy in (YYZ+23; ZGK+24). Following the original paper, we perform offline evaluation focusing on the element accuracy, Operation F1 and step success rate averaged across the task.

In the first section of the table (row 1-3), We report numbers from a set of open source VL models as it appears in (ZGK+24; CSC+24). Here CogAgent and Qwen-VL are not finetuned on the Mind2Web training set. More detailed information about model settings can be found in the Appendix8.4.

In the second section of the table (row 4-9) we report numbers from Mind2web paper (DGZ+23) and SeeAct (ZGK+24) paper. In this section, all of the approaches use the HTML elements selected by a finetuned element proposal model on Mind2Web training set which produces top 50 relevant elements on the HTML page based on the user task. Additionally, GPT-4V+SOM and GPT-4V+textual choices corresponds to the SeeAct with image annotation, and textual choices grounding methods respectively. In GPT-4V+SOM, the set of mark (SOM) boxes are selected from the element proposal model, and are labeled with the ground truth location extracted from HTML. In contrast, GPT-4V+textual uses DOM information of the selected relevant elements directly in the text prompt, rather than overlaying bounding boxes on top of screenshot. The better performance of textual choice corroborates with the experiment results in 4.1.

In the last section (row 10), we report numbers of OMNIPARSER using GPT-4V as the action prediction model. We observe OMNIPARSER achieves comparable performance to model that uses additional HTML information and additionally finetuned element proposal model.

In summary, without using parsed HTML information, OMNIPARSER is able to outperform GPT-4's performance that uses HTML in every sub-category by a significant margin, suggesting the substantial benefit of the screen parsing results provided by OMNIPARSER. Additionally, OMNIPARSER outperforms the GPT-4V+SOM approach by a large margin where the set of marks coordinates are extracted from the html. This suggests that OMNIPARSER's interactable detection model and the icon description model provides useful information to perform the task. Compared to GPT-4V+textual choices, OMNIPARSER significantly outperforms in Cross-Website and Cross-Domain category

(+4.1% and +5.2%), while underperforming (-0.8%) slightly in the Cross-Task category, which indicates that OMNIPARSER provides higher quality information compared to ground truth element information from DOM and top-k relevant elemnt proposal used by the GPT-4V+textual choices set-up, and make the GPT-4V easier to make a accurate action prediction. Lastly, OMNIPARSER with GPT-4V significantly outperform all the other trained models using only UI screenshot such as SeeClick and Qwen-VL.

| Methods | Input Types | | Cross-Website | | | Cross-Domain | | | Cross-Task | | |
|---|---|---|---|---|---|---|---|---|---|---|---|
| | HTML free | image | Ele.Acc | Op.F1 | Step SR | Ele.Acc | Op.F1 | Step SR | Ele.Acc | Op.F1 | Step SR |
| CogAgent | ✓ | ✓ | 18.4 | 42.2 | 13.4 | 20.6 | 42.0 | 15.5 | 22.4 | 53.0 | 17.6 |
| Qwen-VL | ✓ | ✓ | 13.2 | 83.5 | 9.2 | 14.1 | 84.3 | 12.0 | 14.1 | 84.3 | 12.0 |
| SeeClick | ✓ | ✓ | 21.4 | 80.6 | 16.4 | 23.2 | 84.8 | 20.8 | 28.3 | 87.0 | 25.5 |
| MindAct (gen) | ✗ | ✗ | 13.9 | 44.7 | 11.0 | 14.2 | 44.7 | 11.9 | 14.2 | 44.7 | 11.9 |
| MindAct | ✗ | ✗ | **42.0** | 65.2 | **38.9** | 42.1 | 66.5 | 39.6 | 42.1 | 66.5 | 39.6 |
| GPT-3.5-Turbo | ✗ | ✗ | 19.3 | 48.8 | 16.2 | 21.6 | 52.8 | 18.6 | 21.6 | 52.8 | 18.6 |
| GPT-4 | ✗ | ✗ | 35.8 | 51.1 | 30.1 | 37.1 | 46.5 | 26.4 | 41.6 | 60.6 | 36.2 |
| GPT-4V+som | ✗ | ✓ | - | - | 32.7 | - | - | 23.7 | - | - | 20.3 |
| GPT-4V+textual choice | ✗ | ✓ | 38.0 | 67.8 | 32.4 | 42.4 | 69.3 | 36.8 | **46.4** | 73.4 | **40.2** |
| OmniParser (GPT-4V) | ✓ | ✓ | 41.0 | **84.8** | 36.5 | **45.5** | **85.7** | **42.0** | 42.4 | **87.6** | 39.4 |

Table 3: Comparison of different methods across various categories on Mind2Web benchmark.

## 4.4 EVALUATION ON ANDROID-IN-THE-WILD

In additional to evaluation on multi-step web browsing tasks, we assess OMNIPARSER on the mobile navigating benchmark AITW (RLR[+]23), which contains 30k instructions and 715k trajectories. We use the same train/test split as in (CSC[+]24) based on instructions, which retain only one trajectory for each instructions and no intersection between train and test. For a fair comparison, we only use their test split for evaluation and discard the train set as our method does not require finetuing.

In table 4, we report the GPT-4V baseline in (YYZ[+]23) paper, which corresponds to the best performing set up (GPT-4V ZS+history) that uses UI elements detected by IconNet (SWL[+]22) through set-of-marks prompting (YZL[+]23) for each screenshot at every step of the evaluation. The detected UI elements consist of either OCR-detected text or an icon class label, which is one of the 96 possible icon types identified by IconNet. Additionally, action history is also incorporated at each step's prompt as well. We used the exact same prompt format in (YYZ[+]23) except the results from the IconNet model is replaced with the output of the finetuned interactable region detection (ID) model. Interestingly, we found that the ID model can generalize well to mobile screen. By replacing the IconNet with the interactable region detection (ID) model we finetuned on the collected webpages, and incorporating local semantics of icon functionality (LS), we find OMNIPARSER delivers significantly improved performance across most sub-categories, and a 4.7% increase in the overall score compared to the best performing GPT-4V + history baseline.

| Methods | Modality | General | Install | GoogleApps | Single | WebShopping | Overall |
|---|---|---|---|---|---|---|---|
| ChatGPT-CoT | Text | 5.9 | 4.4 | 10.5 | 9.4 | 8.4 | 7.7 |
| PaLM2-CoT | Text | - | - | - | - | - | 39.6 |
| GPT-4V image-only | Image | 41.7 | 42.6 | 49.8 | 72.8 | 45.7 | 50.5 |
| GPT-4V + history | Image | 43.0 | 46.1 | 49.2 | **78.3** | 48.2 | 53.0 |
| OmniParser (GPT-4V) | Image | **48.3** | **57.8** | **51.6** | 77.4 | **52.9** | **57.7** |

Table 4: Comparison of different methods across various tasks and overall performance in AITW benchmark.

## 4.5 EVALUATION ON WINDOWSAGENTARENA

In this section, we report numbers in the concurrent work Windows Agent Arena (BZB[+]24) paper in table 5. Windows Agent Arena is a general environment for Windows operating system (OS) where the models are evaluated on 150+ user tasks in a real Windows OS and use a wide range of applications, tools, and web browsers. For cleaner comparison, we consider the cases that do not use additional UIA tree information from the windows environment. The baseline Pytesseract+DOM+Grounding DINO approach uses Pytesseract python package for OCR, Grounding DINO for icon detection and additionally incorporating DOM tree information. We observe that OmniParser achieves better performance than the baseline by only using screenshot consistently across different models.

| Methods | Models | Office | Web Browser | Windows System | Coding | Media & Video | Windows Utils | Total |
|---|---|---|---|---|---|---|---|---|
| | Phi3-V | 0% | 0% | 4.2% | 4.3% | 0.0% | 0.0% | 1.3% |
| Pytesseract + DOM + | GPT-4o | 0.0% | 0.0% | 29.2% | 0.0% | 5.0% | 0.0% | 5.2% |
| Grounding DINO | GPT-4V-1106 | 0.0% | 10.3% | 21.3% | 12.5% | 9.8% | 0.0% | 8.6% |
| | Phi3-V | 0.0% | 0.0% | 8.6% | 0.0% | 5.0% | 0.0% | 2.0% |
| Omniparser | GPT-4o | 0.0% | 6.7% | 30.3% | 4.3% | 15.3% | 8.3% | 9.4% |
| | GPT-4V-1106 | 2.3% | 23.6% | 20.8% | 8.3% | 20.0% | 0.0% | 12.5% |
| Human performance | - | 75.8% | 76.7% | 83.3% | 68.4% | 42.8% | 91.7% | 74.5% |

Table 5: Comparison of different methods' performance on Windows Agent Arena benchmark.

# 5 ABLATIONS

To further demonstrate OMNIPARSER is a plugin choice for off-the-shelf vision langauge models, we show the performance of OMNIPARSER combined with recently announced vision language models: Phi-3.5-V (AAA⁺24) and Llama-3.2-V (DJP⁺24). As seen in table 6, our finetuned interactable region detection (ID) model significantly improves the task performance compared to grounding dino model (GD) with local semantics across all subcategories for GPT-4V, Phi-3.5-V and Llama-3.2-V. In addition, the local semantics of icon functionality helps significantly with the performance for every vision language model.

| Methods | Models | Model Size | Mobile | Desktop | Web | Average |
|---|---|---|---|---|---|---|
| OmniParser | GPT-4V | - | 75.5% | 77.5% | 66.2% | 73.0% |
| - w.o. ID | GPT-4V | - | 74.3% | 67.1% | 64.1% | 68.7% |
| - w.o. ID and w.o LS | GPT-4V | - | 71.1% | 45.6% | 58.5% | 58.4% |
| OmniParser | Phi-3.5-V | 4.2B | 39.4% | 39.2% | 24.0% | 34.2% |
| - w.o. ID | Phi-3.5-V | 4.2B | 38.1% | 32.5% | 22.2% | 30.9% |
| - w.o. ID and w.o LS | Phi-3.5-V | 4.2B | 32.9% | 31.0% | 18.5% | 27.5% |
| OmniParser | Llama-3.2-V | 11B | 47.6% | 48.1% | 37.4% | 44.4% |
| - w.o. ID | Llama-3.2-V | 11B | 45.7% | 44.6% | 37.5% | 42.6% |
| - w.o. ID and w.o LS | Llama-3.2-V | 11B | 38.5% | 37.3% | 31.2% | 35.6% |

Table 6: Ablation study of OMNIPARSER performance using different vision language models on ScreenSpot Benchmark. LS is short for local semantics of icon functionality, ID is short for the interactable region detection model we finetune. The setting w.o. ID means we replace the ID model with original Grounding DINO model not finetuned on our data, and with local semantics. The setting w.o. ID and w.o LS means we use Grounding DINO model, and further without using the icon description in the text prompt.

# 6 DISCUSSIONS

In this section, we discuss a couple of common failure cases of OMNIPARSER with examples and potential approach to improve.

**Repeated Icons/Texts** From analysis of the the GPT-4V's response log, we found that GPT-4V often fails to make the correct prediction when the results of the OMNIPARSER contains multiple repeated icons/texts, which will lead to failure if the user task requires clicking on one of the buttons. This is illustrated by the figure 8 (Left) in the Appendix. A potential solution to this is to add finer grain descriptions to the repeated elements in the UI screenshot, so that the GPT-4V is aware of the existence of repeated elements and take it into account when predicting next action.

**Corase Prediction of Bounding Boxes** One common failure case of OMNIPARSER is that it fails to detect the bounding boxes with correct granularity. In figure 8 (Right), the task is to click on the text 'MORE'. The OCR module of OMNIPARSER detects text bounding box 8 which encompass the desired text. But since it uses center of the box as predicted click point, it falls outside of the ground truth bounding box. This is essentially due to the fact that the OCR module we use does not have a notion of which text region are hyperlink and clickable. Hence we plan to train a model that combines OCR and interactable region detection into one module so that it can better detect the clickable text/hyperlinks.

**Icon Misinterpretation** We found that in some cases the icon with similar shape can have different meanings depending on the UI screenshot. For example, in figure 9, the task is to find button related

to 'More information', where the ground truth is to click the three dots icon in the upper right part of the screenshot. OMNIPARSER successfully detects all the relevant bounding boxes, but the icon description model interpret it as: "a loading or buffering indicator". We think this is due to the fact that the icon description model is only able to see each icon cropped from image, while not able to see the whole picture during both training and inference. So without knowing the full context of the image, a symbol of three dots can indeed mean loading buffer in other scenarios. A potential fix to this is to train an icon description model that is aware of the full context of the image.

## 7 CONCLUSION

In this paper, We propose OMNIPARSER, a general vision only approach that parse UI screenshots into structured elements. OMNIPARSER encompasses two finetuned models: an icon detection model and a functional description models. To train them, we curated an interactable region detection dataset using popular webpages, and an icon functional description dataset. We demonstrate that with the parsed results, the performance of GPT-4V is greatly improved on ScreenSpot benchmarks. It achieves better performance compared to GPT-4V agent that uses HTML extracted information on Mind2Web, and outperforms GPT-4V augmented with specialized Android icon detection model on AITW benchmark. We hope OMNIPARSER can serve as a general and easy-to-use tool that has the capability to parse general user screen across both PC and mobile platforms without any dependency on extra information such as HTML and view hierarchy in Android.

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

# 8 APPENDIX

## 8.1 DETAILS OF ICON-DESCRIPTION DATASET

In figure 5, we see that the original BLIP-2 model tend to focus on describing shapes and colors of app icons, while struggling to recognize the semantics of the icon. This motivates us to finetune this model on an icon description dataset. For the dataset, we use the result of parsed icon bounding boxes inferenced by the interactable icon detection model on the ScreenSpot dataset since it contains screenshots on both mobile and PC. For the description, we ask GPT-4o whether the object presented in the parsed bounding box is an app icon. If GPT-4o decides the image is an icon, it outputs one-sentence description of the icon about the potential functionality. And if not, GPT-4o will output 'this is not an icon', while still including this in the dataset. In the end, we collected 7185 icon-description pairs for finetuning.

We manually inspected the dataset to include icon-description pairs across a wide range of functions. These icons include system icons and popular software/app icons. After deduplication, we have 174 app icons in the pc platform, and 170 app icons in mobile platform. Further, we leveraged GPT-4o and conducted an analysis of the distribution of the icons. We first summarize the icons into the following types:

1. **Functional Icons.** These icons represent actions or functionalities users can perform. Subcategory:
   - **Navigation Icons:** Back, forward, home, refresh, menu.
   - **Action Icons:** Add (+), delete (trash), edit (pencil), search (magnifying glass), share, upload, download.
   - **System Actions:** Lock, log out, power off, settings (gear icon).
2. **Informational Icons.** These icons convey information or statuses. Examples include: Subcategory:
   - **Notification Icons:** Alerts, messages, updates.
   - **Status Icons:** Battery level, network signal, Wi-Fi, Bluetooth, processing/loading (spinner).
   - **Error or Warning Icons:** Exclamation marks, red crosses, or triangles.
3. **App-Specific Icons.** These icons are unique to a specific app or service.
4. **Media Control Icons.** These icons control media playback. Examples include play, pause, stop, fast forward, rewind, volume up, volume down.

The detail distribution of each sub-category is presented in figure 4. With this dataset, the resulting model demonstrates strong generalization on varied benchmarks and real-world applications.

We finetune BLIP-2 model for 1 epoch on the generated dataset with constant learning rate of $1e^{-5}$, no weight decay and Adam optimizer. We show a few of the qualitative examples of finetuned model vs the original model in figure 5.

## 8.2 TRAINING DETAILS OF INTERACTABLE ICON REGION DETECTION MODEL

As introduced in 3.1, we train a YOLOv8 model on the interactable icon region detection dataset. We collect in total of 66990 samples where we split 95% (63641) for training, and 5% (3349) for validation. We train for 20 epochs with batch size of 256, learning rate of $1e^{-3}$, and the Adam optimizer on 4 GPUs. We show the training curve in figure 6.

## 8.3 DETAILS OF SEEASSIGN EVALUATION

### 8.3.1 PROMPT USED FOR GPT-4V

GPT-4V without local semantics:

```
Here is a UI screenshot image with bounding boxes and corresponding
    labeled ID overlaid on top of it, your task is {task}. Which icon
    box label you should operate on? Give a brief analysis, then put your
     answer in the format of \n```Box with label ID: [xx]```\n
```

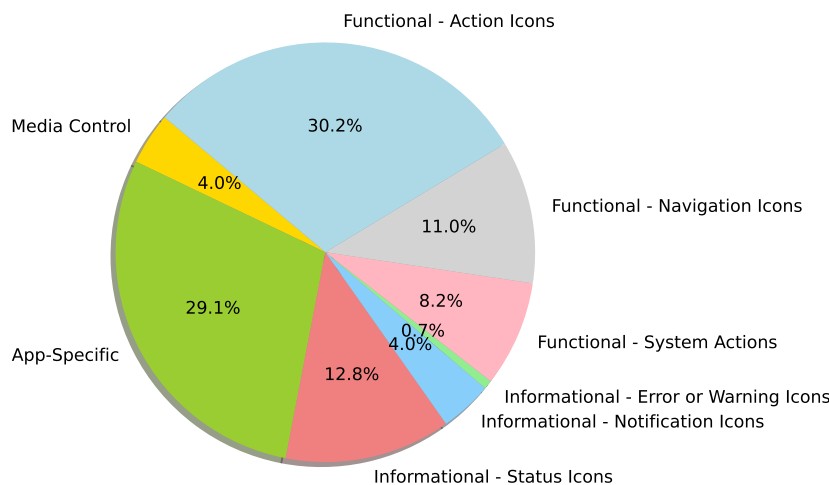

Figure 4: Distribution of the collected icons in the icon description dataset

| Before | | After |
|---|---|---|
| | an app icon with a pie chart on it | a presentation or screen sharing application |
| | the microsoft office logo is shown in a circle | Microsoft Outlook, an email application. |
| | an iphone app with an image of a flower | Photos, a photo-sharing application. |
| | an orange and white logo with a smiley face | Discord, a messaging and voice chat application. |
| | a blue app icon with a person on it | a location or location-related function. |
| | a grey and white image of a gear wheel | Settings. |

Figure 5: Example comparisons of icon description model using BLIP-2 (Left) and its finetuned version (Right). Original BLIP-2 model tend to focus on describing shapes and colors of app icons. After finetuning on the functionality semantics dataset, the model is able to show understanding of semantics of some common app icons.

GPT-4V with local semantics:

```
Here is a UI screenshot image with bounding boxes and corresponding
    labeled ID overlayed on top of it, and here is a list of icon/text
    box description: {parsed_local_semantics}. Your task is {task}. Which
     bounding box label you should operate on? Give a brief analysis,
    then put your answer in the format of \n'''Box with label ID: [xx]'''\
    n
```

## 8.4 DETAILS OF MIND2WEB EVALUATION

Here we list more details of each baseline in table 3.

**SeeClick, QWen-VL** SeeClick is a finetuned version of Qwen-VL on the Mind2Web training set and we report both of their numbers in their paper.

**CogAgent** CogAgent number is taken from the SEEAct paper (ZGK[+]24), where they report cogagent-chat-hf checkpoint that is not fine-tuned on Mind2Web for experiments.

**MindAct(Gen), MindAct, GPT-3.5-Turbo, GPT-4** The numbers for these baseline are taken from

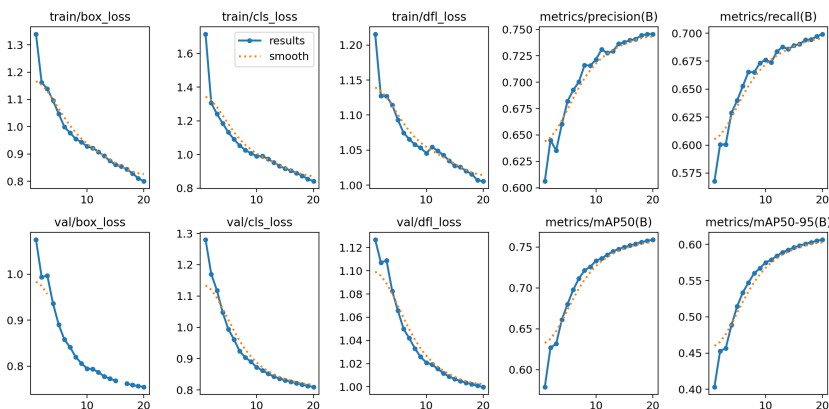

Figure 6: Training curves of interactable icon region detection model.

the Mind2Web (DGZ+23) paper, where they use HTML information to augument the corresponding web agent.

**GPT-4V+som** This model corresponds to the image annotation grounding method in SeeAct paper, where the som boxes extracted from the selelcted HTML elements are provided to GPT-4V to make action prediction.

**GPT-4V+textual choice** This corresponds to the best performing scenario in SeeAct paper (except the Oracle), that uses the selected HTML elments information in a multi-choice question format as input to the GPT-4V agent.

### 8.4.1 QUALITATIVE EXAMPLES

We list a few more examples to demonstrate local semantics of icon function description helps GPT-4V make better action prediction in figure 7.

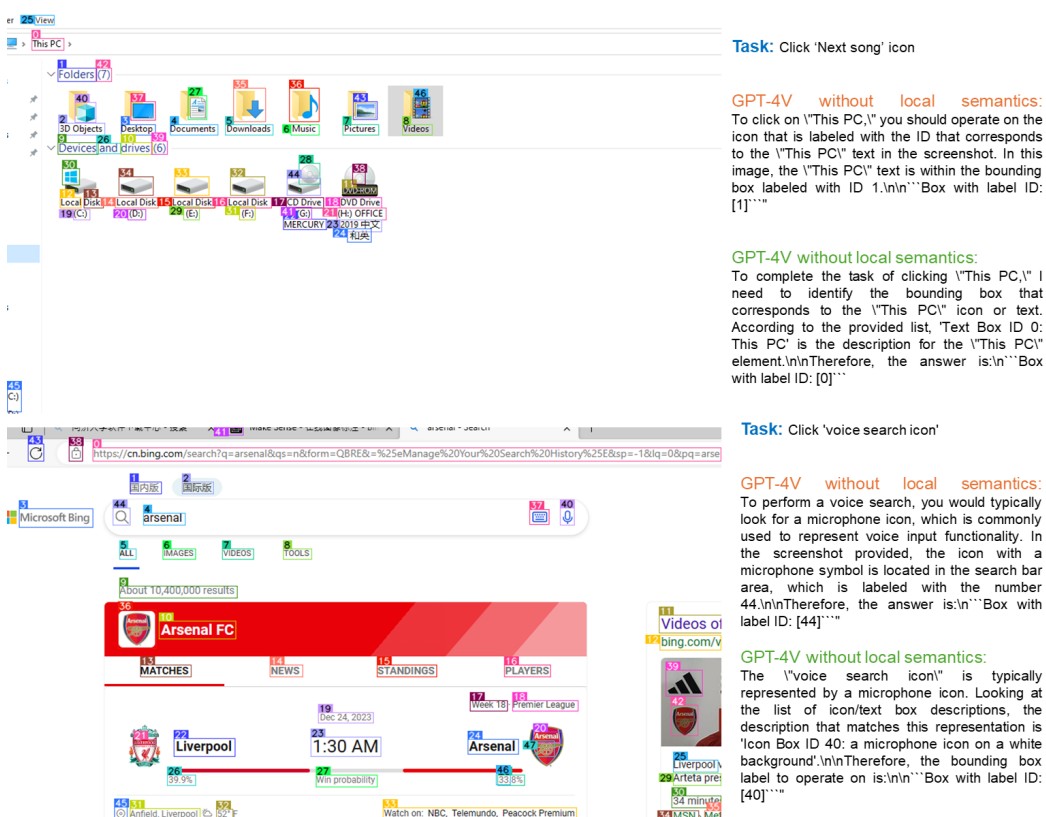

Figure 7: More examples of local semantics of icon functionality help with GPT-4V in grounding actions

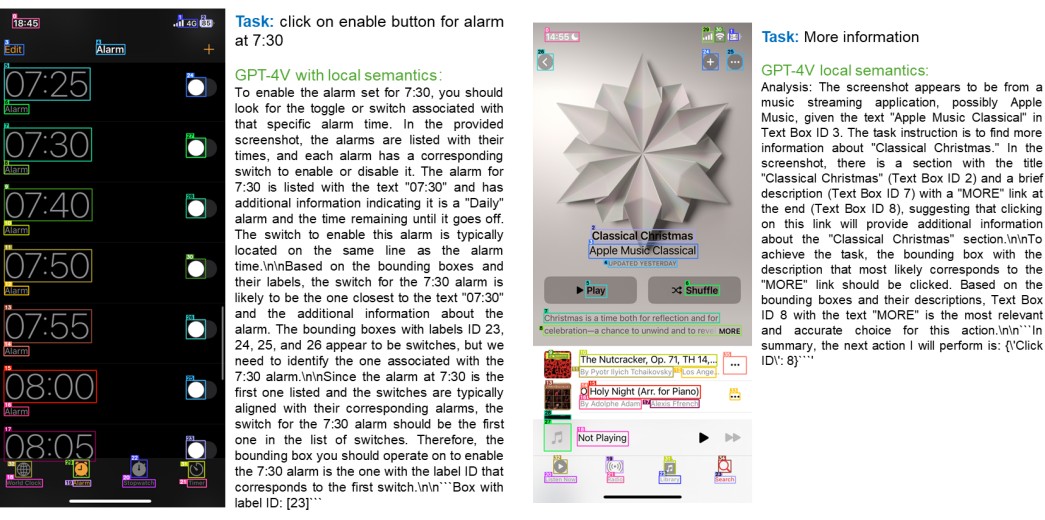

Figure 8: Analysis of failure cases. All the bounding boxes are labeled by which relies only on the screenshot. **Left**: There are in total 7 similar enable button for 7 different alarm times in the parsed screenshot. And the correct Icon ID corresponding to alarm 7:30 is 27. GPT-4V fails to make the correct prediction. **Right:** The ground truth region to click is the text 'MORE' inside bounding box 8. We can see that the OCR fails to detect the text 'MORE' in bold, and only detects the bounding box 8, which encompasses 'MORE'. Since the predicts the click point as the center of the box, so it the predicted click point falls outside of the ground truth region, which leads to failure in this task.

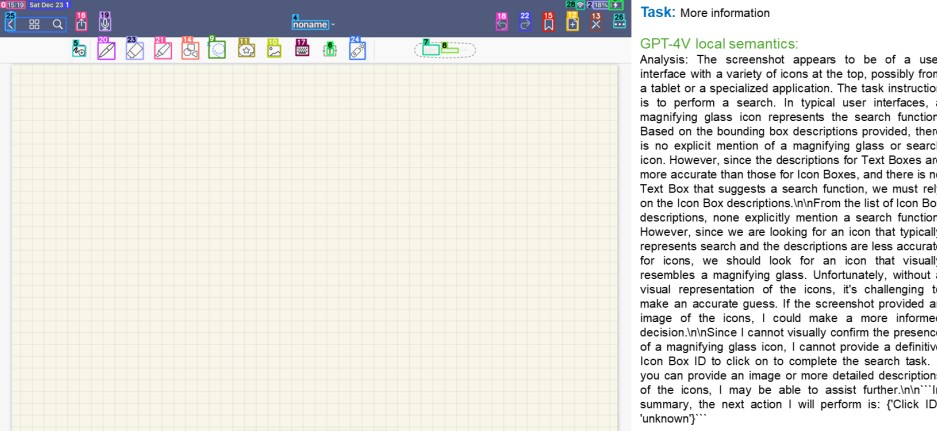

Figure 9: Analysis of failure cases. The task is to find button related to 'More information', and the ground truth is to click the three dots icon in the upper right part of the screenshot. The the icon functional description model does not take into account the context of this page and interpret it as: "a loading or buffering indicator" which causes the failure.

