# OpenReview forum: "OmniParser for Pure Vision Based GUI Agent"
_ICLR.cc/2025/Conference — Submitted to ICLR 2025_

### Official Review · Reviewer_HZMo · 2024-10-28

**Soundness:** 3
**Presentation:** 3
**Contribution:** 3
**Rating:** 3
**Confidence:** 5

**Summary:**

The paper introduces Ominiparser, a modular method designed to enhance large vision-language models e.g., GPT-4V, specifically for parsing user interface (UI) screenshots and grounding abilities to screen elements effectively.
Ominiparser addresses current limitations in screen parsing, especially the inability of existing models to detect interactable regions and associate actions with specific areas in a UI.
To achieve this, the authors:
1. curate a dataset for interactable icon detection and icon description from DOM tree;
2. subsequently using them to fine-tune models including a captioner BLIP-2 and a detector YOLO, served as a combination solution.
3. Ominiparser is tested on benchmarks like ScreenSpot, Mind2Web, and AITW, which connects with an advanced planner, show substantial improvements over baseline models (e.g., GPT-4V), enabling process screenshots alone without relying on additional HTML or view hierarchy data.

**Strengths:**

1. This work focuses on a critical problem in UI vision understanding: recognizing elements solely from visual data, i.e., screenshots, rather than relying on DOM data.

2. The proposed strategy is flexible and can be applied across multiple planning frameworks.

3. The grounding data takes into account the functionality of the icon and has a high-level understanding.

4. The experiment has an online setting on Window Agent Arena, which is helpful.

**Weaknesses:**

1. While this work provides great practical contributions, the main focus is on its novelty and innovation; it mainly involves fine-tuning existing models using new data sources and applying these models through ensemble methods. The existing innovations do not meet the criteria for ICLR submissions, but are suitable for submission in technical reports or industry workshops.

2. There is limited analysis and discussion regarding UI visual understanding across platforms or data sources. For instance, while the community has substantial OCR data related to UI, icon-specific data remains scarce. The paper does not include distribution details on the proposed data, such as the number or sources of icons and software applications represented.

3. The authors do not provide analysis on the impact of prompt length when combining data from different sources (e.g., GPT-4, GPT-4V, with or without additional tool-generated information). Additionally, there is no breakdown of the financial cost or computation time associated with using GPT-4V alongside Omniparser to complete individual tasks.

4. The paper does not propose an innovative strategy for handling UI visual understanding with models like BLIP-2 or YOLO, particularly given that UI screenshots are often large (e.g., 2K resolution). Efficiency strategies could enhance processing in such high-resolution contexts.

**Questions:**

Small issue -- Typo:
line 837 -- QWen -> Qwen

- Instead of applying both BLIP-2 or YOLO, how about train an open-world detector that let GPT4 to determine which element we should ground and output a text query, then use this query for object grounding?

- would the model, training and benchmark data will be open-source?

**Details Of Ethics Concerns:**

The Ethics lie in the training data collection. The author should provide a clarification about this part.

---

### Official Review · Reviewer_MNEC · 2024-11-03

**Soundness:** 3
**Presentation:** 3
**Contribution:** 3
**Rating:** 5
**Confidence:** 4

**Summary:**

This paper introduces OmniParser, a vision-based GUI agent designed to enhance large vision-language models (such as GPT-4V) by effectively parsing user interface (UI) screenshots. OmniParser comprises several key components: an OCR module for extracting text elements, an icon detection model for identifying interactable icons, a captioning model to describe the functions of detected elements, and a generalist large vision-language model for interpreting the parsed information to make reasoning and action decisions. The authors curated datasets to train the icon detection and captioning models, fine-tuning them to improve their performance. They tested OmniParser on benchmarks such as ScreenSpot, Mind2Web, and AITW, demonstrating that the proposed framework significantly enhances the capabilities of vision-based agents over baseline models by enabling them to process screenshots without relying on textual inputs.

**Strengths:**

1. This paper is well written and easy to follow. The demonstration of the GUI framework is clear.
2. The proposed strategy, which involves extracting the position of interactable elements along with their function descriptions, is both flexible and effective. It significantly enhances GPT-4V’s agentic capabilities, particularly the GUI grounding capabilities of the entire framework.
3. The data curation considers the position and function description of interactable elements, which would be beneficial to the research community if it were to be open-sourced.

**Weaknesses:**

1. The novelty and innovation in this work are limited. The primary approach involves training existing modules for various purposes, such as icon detection and captioning, and then integrating these modules to construct a GUI framework. This does not meet the criteria for ICLR.

2. Missing details.

    - 2.1 The authors collected data for interactable region detection from web pages. However, in Figure 2, most examples contain text elements, with very few icons shown. If most of the interactable elements from web pages are texts, how does the model generalize to icons from other domains, such as mobile and desktop interfaces? Additionally, what are the categories of interactable elements and their distributions in the training data?

    - 2.2. The authors curate the icon description from the ScreenSpot dataset, since the data scale of ScreenSpot is not large, how do the authors guarantee the model’s generalization to other datasets, benchmarks or real-world applications?

     - 2.3 Evaluation from the cost perspective: Since OmniParser combines different modules for OCR, element detection, element description, and action decision, the inference time could be longer when compared to end-to-end frameworks, such as SeeClick[1], or more advanced VLMs (e.g., Qwen2-VL) trained with SeeClick data. Additionally, OmniParser utilizes GPT-4V in its framework, but the associated cost in USD is not included in the analysis.

[1] Cheng K, Sun Q, Chu Y, et al. Seeclick: Harnessing gui grounding for advanced visual gui agents[J]. arXiv preprint arXiv:2401.10935, 2024.

**Questions:**

1. typos: l198 - l199: ‘click on ‘settings”, ‘click on the minimize button’ → “click on settings”, “click on the minimize button”
2. Related questions from the weakness part.
3. Are the authors going to release the following? (1) the icon detection and description dataset, (2) the trained models, (3) code for data collection.

---

### Official Review · Reviewer_DUX7 · 2024-11-03

**Soundness:** 3
**Presentation:** 3
**Contribution:** 3
**Rating:** 6
**Confidence:** 5

**Summary:**

The paper introduces OMNIPARSER, a universal UI interface parsing methodology that addresses the challenges of reliably identifying interactive elements in user interfaces. By developing a dataset based on popular webpage DOM trees for interactive area detection and an icon-description dataset, the approach trains task-specific models to understand the semantics of elements within screenshots and accurately map intended actions to corresponding screen regions. OMNIPARSER, through its integration of multiple fine-tuned models, achieves enhanced screen understanding capabilities and demonstrates significant improvements across benchmarks including ScreenSpot, Mind2Web, and AITW, validating its seamless compatibility with existing vision-language models.

**Strengths:**

1. The authors conducted comprehensive evaluations on multiple benchmark datasets, with OmniParser achieving state-of-the-art results across all datasets.

2. OmniParser does not rely on additional information like DOM or view hierarchy, making it more generalizable.

3. The authors performed extensive ablation studies validating the effectiveness of the proposed ID and IS modules, and verified the robustness of the entire framework on open-source Llama-3.2-V and Phi-3.5-V models.

4. The authors released their code and models, making it easy for researchers to reproduce the results.

**Weaknesses:**

1. OmniParser's strong performance heavily relies on the powerful backbone model (GPT4-V), and switching to open-source models would significantly decrease its performance.

2. The entire pipeline is not end-to-end, which increases its complexity and inference latency.

**Questions:**

1. If GPT-4V can locate objects without relying on SOM in the future, would this pipeline still be effective?

2. Would better results be achieved by training an end-to-end model directly with constructed data, rather than relying on additional fine-tuned models?

---

### Official Review · Reviewer_ZAy2 · 2024-11-03

**Soundness:** 3
**Presentation:** 3
**Contribution:** 3
**Rating:** 6
**Confidence:** 5

**Summary:**

This paper introduces OMNIPARSER, a method designed to enhance the action-generating capabilities of multimodal models like GPT-4V when interacting with user interfaces. Specifically, the paper collects an interactable icon detection dataset with 67,000 unique screenshot images and DOM annotations. Then, the YOLO-v8 model is fine-tuned to detect icons within a screenshot. Additionally, an icon description dataset with 7,185 icon descriptions is built, and BLIP v2 is fine-tuned on it to output icon functionalities. Consequently, given a screenshot, OMNIPARSER can detect UI elements through the proposed icon detection, caption, and OCR models. It can further incorporate GPT-4V to generate GUI actions. Experiments on Mind2Web and AITW demonstrate promising improvements over the baselines.

**Strengths:**

- The paper proposes an alternative method for parsing GUI elements in screenshots by using an icon detection model combined with a description model to generate comprehensive information about the elements.
- The collected dataset on icon detection and description is expected to be highly beneficial for the community.
- The experiments are rich and thorough. And the benchmark performance shows promising results.

**Weaknesses:**

1. Since the paper mainly uses existing YOLO-v8 and BLIP v2 models, its primary contribution lies in the proposed icon-detection and description datasets. However, many details regarding dataset construction and data statistics are missing.

   - The authors mention that the main focus is on collecting an "INTERACTABLE REGION DETECTION" dataset. However, how are elements deemed interactive? The DOM does not directly provide metadata indicating interactivity, only element types. How was this implemented, and does this process have a risk of misclassification?

   - For the icon description dataset, the authors used GPT-4o for labeling. However, GPT-4o itself has limitations in icon recognition. What mechanisms did the authors employ to ensure data quality, and what is the accuracy of the generated descriptions?

   - The authors mention annotating 7,185 icon-description data points. Are all these icons unique? Could the authors provide data distribution examples, including icon types?

2. The authors mention merging DOM and OCR results by calculating overlap and merging bounding boxes with high overlap (>90%). For web applications, does this approach introduce substantial redundancy? For instance, in the bottom-left image of Figure 2, the bounding box for "Contact Us" is likely derived from the DOM and should generate a larger box, while OCR might produce a tighter one with lower overlap. More examples demonstrating the effectiveness of the merging strategy would be helpful.

3. Some comparisons appear unfair. For instance, using OCR predictions for grounding in ScreenSpot will likely outperform existing MLLMs, but this does not necessarily highlight the model’s contribution. Similarly, in results on other benchmarks, it is difficult to determine whether OCR results or icon detection contribute more. Testing GPT-4V + OCR to demonstrate any performance boost would be beneficial.

**Questions:**

- What is the OCR module used in the Omniparser?

- Since each icon requires a description generated by BLIP-2, could this contribute to increased model latency?

- What are the performance levels of the separate icon detection and description modules, and which one currently presents a bottleneck? Additionally, it seems that no validation set is included for the icon description at this stage. It is quite hard to control when to stop training.

- Since GPT-4V can perform icon description directly, would it be feasible to test its performance on description tasks using GPT-4V?

---

### Meta-Review · Area_Chair_A6NC · 2024-12-20

**Metareview:**

This paper is borderline with two reviewers slightly positive, and two reviewers on the negative side. Overall, the reviewers generally agree that the paper is well-written and the proposed method shows promising results on various benchmarks. However, they raise concerns about the novelty and complexity of the approach. More specifically, the reviewers liked strong empirical results that Omniparser shows on multiple benchmarks. Omniparser seems also be able to generalize to work with different LVLMs and across various platforms. The reviewers also liked the dataset contribution. On the other hand, the reviewers are concerned with the limited novelty of the work and feel the work doesn't meet the bar of ICLR. The pipeline is also complex, which seems an integration of existing models.

Based on the reviews and discussion, the paper appears to fall short of the acceptance threshold. The lack of fundamental technical novelty and the complexity of the pipeline outweigh the empirical results and dataset contribution. Addressing these concerns can significantly improve the contribution of the work.

**Additional Comments On Reviewer Discussion:**

Re: Novelty: The authors argue that their main contribution lies in identifying and addressing the limitations of existing LVLMs in UI understanding. They also highlight the practicality and efficiency of their approach compared to training end-to-end LVLMs.
Re: Data: Reviewers raise concerns about the details and generalization of the datasets. The authors provide additional information and analysis to address these concerns.
Re: Latency and Cost: The authors acknowledge the latency issue and plan to explore optimizations in future work. They also provide an analysis of the cost associated with using GPT-4V.
Overall, reviewers are not moved by the rebuttal.

---

### Decision · Program_Chairs · 2025-01-22

Reject